Experiences of international medical students enrolled in Chinese medical institutions towards online teaching during the COVID-19 pandemic

http://orcid.org/0000-0001-7414-7572 Aslam Sarfraz 1 sarfrazmian@nenu.edu.cn
Akram Huma 2
http://orcid.org/0000-0002-7595-842X Saleem Atif 3
Zhang BaoHui 1 baohui.zhang@snnu.edu.cn
1 School of Education, Shaanxi Normal University , Xi’an, Shaanxi , China
2 Faculty of Education, Northeast Normal University , Changchun, Jilin , China
3 College of Teacher Education, College of Education and Human Development, Zhejiang Normal University , Jinhua, Zhejiang , China
Suner Aslı
Electronic publication date: 2021 Aug 25
Publication date: 2021
Volume: 9
Electronic Location ID: e12061
Received 2021 May 26; Accepted 2021 Aug 4
Copyright: © 2021 Aslam et al.
Copyright year: 2021
Copyright holder: Aslam et al.
License: This is an open access article distributed under the terms of the Creative Commons Attribution License, which permits unrestricted use, distribution, reproduction and adaptation in any medium and for any purpose provided that it is properly attributed. For attribution, the original author(s), title, publication source (PeerJ) and either DOI or URL of the article must be cited.
License URL: https://creativecommons.org/licenses/by/4.0/

Keywords: Medical students, Medical education, COVID-19, Online teaching, China, Pandemic, International students

Funding: The authors received no funding for this work.

==============================
Introduction

. The COVID-19 pandemic has forced the world to pause. One hundred and eighty-eight countries have imposed countrywide school closures, affecting more than 1.5 billion children and youths. The majority of academic leaders are currently encouraging online education to resolve this crisis. This study aimed to investigate international medical students’ (IMS) experiences of online teaching during the COVID-19 pandemic.

Methods

Data were collected online using a validated questionnaire and one open-ended question, presented on the Google forms platform. The study attracted responses from 1,107 IMS volunteer participants. IBM SPSS v. 25, GraphPad Prism v. 9, and MindManager v. 2018 were used for data analysis. All variables were subjected to descriptive statistical analysis. The Mann–Whitney U test was used in subgroup analysis and the Kruskal-Wallis test was also applied for year-wise comparisons. Open-ended text responses were analyzed qualitatively, extracting themes by which responses were classified.

Results

Among 1,107 respondents, a total of 67.8% were males, and the majority (63.1%) of the IMS were in the age group of 21–23 years. The results show that more than half of the respondents reported their Internet connection quality as poor to average. Poor Internet connection severely affected IMS online learning experience. Persistent and recurrent issues with Internet access became a significant concern for IMS. Lack of electricity is one of the factors that can contribute to poor learning output and dissatisfaction with online teaching. IMS perceive online medical education as unhelpful in several phases of the training, such as improving their clinical skills, knowledge, and discussion skills.

Conclusions

During these unprecedented periods, online teaching has allowed medical education to continue. However, IMS are generally dissatisfied with online teaching. Medical students must visualize the human body, so supportive technologies are important to compensate for the lack of clinical practices. Medical institutions may need to invest in faculty training programs and continually adjust to enhance the content of online training and international partnerships. A switch from conventional face-to-face teaching to a fully functional virtual education framework in the medical education field will take time and experience.

Introduction

Current scenario: the world and China

The twenty-first century is experiencing what may be one of its most devastating events. Now known to the world as the COVID-19 pandemic, the virus swiftly engulfed the whole world with almost 11 million cases in a span of around six months. It has not only increased the global burden of disease but has heavily dented many social systems, including education (Baloch et al., 2021). The World Health Organization (WHO) announced the COVID-19 outbreak initially as a public health emergency of international concern on January 30, 2020 and later declared it a pandemic on March 11, 2020 (WHO, 2020). In China, the first case of novel coronavirus was reported in Wuhan in December 2019 (Chen et al., 2020; Huang et al., 2020). At that time it was not expected to become a pandemic, but by June 20th 2020 COVID-19 had infected more than 8.5 million and killed 460 thousand people globally (Sindiani et al., 2020).

Soon after this viral outbreak, semester break was approaching in China and many international students were returning to their homes for vacations. The Chinese Ministry of Foreign Affairs announced that all students (local and international) must await official permission before returning to their institutions (Wang & Dai, 2020). On 28th March 2020, in view of the rapid global spread of COVID-19, China temporarily suspended the right of foreign nationals holding valid visas or residence permits, including international students, to enter the country (Overseas Security Advisory Council, 2020).

With the growing influence of Chinese education, the number of students in China is increasing (Gu et al., 2020). In 2017, China became the leading destination for international students in Asia (Jianfeng, 2018). China is currently one of the fastest-growing destinations for international medical students (IMS) (Fan, Kosik & Chen, 2013), having received over 68,000 IMS, mostly from Asian and African nations (Li, 2019). More than 10,000 students came to China to study medicine in 2018, most of whom chose a Bachelor of Medicine Bachelor of Surgery (MBBS) program delivered in English. Chinese medical institutions offer an English-taught 6-year undergraduate program (MBBS) that includes 5-year theoretical and practical studies courses and a one-year clinical rotation internship. Successful students receive a bachelor’s degree in medicine and surgery at the end of the program. The Chinese government has authorized 49 medical institutions to accept IMS (Chu et al., 2019). Each of those students requires a study visa and many of those planning to work in China after graduation are required to apply for a work visa (Li & Sun, 2019).

The COVID-19 pandemic has forced the world to pause, and countries worldwide have implemented strict policies to avoid disease transmission (Sarwar et al., 2020). It was observed by UNESCO (2020) that 91% of the total global student population has been absent from school in more than 188 countries affected by the pandemic (Koçoglu & Tekdal, 2020).

COVID-19 and online medical education

In order to continue to offer higher education, authorities worldwide issued new recommendations on conversion to online university teaching and most academic leaders are currently encouraging this switch (UNESCO, 2020). In the COVID-19 pandemic situation, universities at all levels worldwide have led their teachers and students to use material conventionally delivered face to face via an online format.

Accelerated development of IT systems and enhancement of Internet mechanisms have allowed online learning to become central to modern global education (Wang, Zhang & Ye, 2020). Moving from on-campus to distance learning may be facilitated by methods such as self-paced independent study and remote interactive workshops, or real-time immersive settings that are needed for distance learning (Cook et al., 2010).

A rise in external resources and teaching programs such as Osmosis and BiteMedicine has allowed many teaching sessions to be available to medical students online (Dost et al., 2020). Learning health and medicine with Osmosis is intended to be fun, with a visual style to help communicate difficult concepts by grounding them with visual memory anchors, memorable characters, and engaging animations (Osmosis, https://www.osmosis.org/about), and BiteMedicine is a free complete resource intended to help medical students excel in their medical studies. Question banks, online textbooks, live webinars and forums are provided to help students to pass their exams (BiteMedicine, https://www.bitemedicine.com).

Several studies indicate that online and blended educational approaches are equivalent to conventional classroom models. However, few studies are based on students and teachers’ satisfaction with online education during situations such as the COVID-19 pandemic (Muflih et al., 2020). Appraising a student’s simulated learning experience may help assess the effectiveness of an online training program (Hamutoglu et al., 2020).

Medical students’ perceptions about online learning

The online learning experiences of medical students globally have been the focus of many recent studies. A national cross-sectional survey (Dost et al., 2020) investigated 2721 UK medical students’ perceptions of online teaching during the COVID-19 pandemic. It was concluded that online teaching had allowed the continuity of medical education during these exceptional times. A cross-sectional survey among medical students in the North of Jordan (Sindiani et al., 2020) found that most medical students favored the conventional face-to-face teaching method over the solo online teaching methods.

A survey of Pakistani dentistry graduates found that they were unanimously unhappy with different online teaching sessions (Sarwar et al., 2020). However, a nationwide survey of online teaching strategies in dental education in China found the online delivery to be necessary and effective during the outbreak. The study recommends that the online education model and pedagogy may be enhanced for future delivery of dental education (Wang, Zhang & Ye, 2020).

As mentioned earlier, China is host to many IMS, but to date no study has investigated their online learning experiences. Research on this issue is important as a basis for development of sophisticated online learning-enabled programs. The present study therefore aimed to: Assess the experiences of IMS regarding the effectiveness of online teaching.

Investigate the challenges faced by IMS in adjusting to this new mode of learning.

Propose practical strategies for medical institutions to address the identified factors.

The findings of this research will serve as the foundation for future applied and intervention studies, as a guide for universities and policymakers worldwide, and may be used to better understand the positive role of online teaching in medical education.

Materials and Methods

A cross-sectional survey design was used for data collection (Fraenkel, Wallen & Hyun, 1993) between January and March 2021 via an online Google-based questionnaire.

Participants

The target population was IMS studying in various medical institutions in China. Undergraduate IMS who were enrolled and took online courses from the first to the fifth year in the 6-year MBBS program were included in this study. A total of 1,107 IMS from fifteen medical institutions across China were recruited by simple random sampling in which each member of the population has an equal chance of being selected. This approach removes bias from the selection procedure and should result in a representative sample (Gravetter & Forzano, 2011). Ideally, a sample size of more than a few hundred is required in order to apply this sampling technique (Saunders, Lewis & Thornhill, 2009). The present study sample size was determined using an Open-Epi online calculator. If 50% of the target population subjects were interested in participating, a sample size of 385 would be required to assess the estimated proportion at 5% absolute precision and 95% confidence (Dhand & Khatkar, 2014). Moreover, a sample size of about 400 should be sufficient for a large population (Krejcie & Morgan, 1970). The G*Power 3.1 calculator was used to compute statistical power of 0.95, above the value of 0.80 considered adequate in social science research (Hair et al., 2016; Uttley, 2019).

Instrument

The data collection tool consisted of two parts: a questionnaire and an open-ended question.

Questionnaire

A validated questionnaire (Sarwar et al., 2020) in English consisting of 31 items (without sub-dimensions or inverse items), with a mixture of question styles including 5-point Likert-type questions, was used to collect the data (File S1). The questionnaire is considered highly reliable, the Cronbach’s alpha value of the original version being 0.78 (Sarwar et al., 2020). The current authors checked validation prior to the final data collection and found Cronbach’s alpha value of 0.83 indicating an acceptable level of internal consistency. This questionnaire was designed and used to measure the self-reported effectiveness of medical e-learning classes during the COVID-19 pandemic. The questionnaire explored the following three themes (1) Demographics of participants (2) General information about technology readiness and online classes (3) Effectiveness of online classes.

Opinions about online teaching

The current researchers added one open-ended question asking ‘‘Please tell us the three most crucial improvements required to make online sessions more effective and anything you want to share; please feel free to share’’ intended to gather IMS perspectives about the challenges they face during online teaching and their suggestions for improvements to make online sessions more effective.

Ethical considerations and participation

Ethics committee approval was received from Shaanxi Normal University’s institutional review board (reference number AR 2021-01-001). IMS were given brief information about the study and were invited to agree to participate using the consent form on the first page of the online questionnaire. Participation was voluntary, and before beginning the survey, participants were told that all data obtained would be anonymous and would be used for research purposes only. At the start of the survey, a mandatory email id was required to validate participation, affirming consent and preventing multiple responses.

Data collection

The survey was created using Google forms, and the link was distributed to IMS globally via social media (WeChat, WhatsApp, Facebook and others). The authors also contacted some known IMS and invited them to let others know about this study to minimize non-response bias. Response rate could not be determined since the number of IMS who were aware of the study was unknown.

Data analysis

Data were exported from Google Forms for data analysis using IBM SPSS v. 25. GraphPad Prism v. 9 was used to generate graphs, and MindManager v. 2018 was used to draw the coding map. Shapiro–Wilk and Kolmogorov–Smirnov normality tests were used to examine whether the data were distributed normally, which found the data set to be non-Gaussian in distribution. Descriptive statistical analysis was performed on all the variables. Mann–Whitney U tests were used in subgroup comparisons between public and private sector and male and female IMS. Kruskal–Wallis tests followed by Mann–Whitney U tests were used for year-wise comparisons. P values of < 0.05 were considered statistically significant.

Content analysis, a qualitative data analysis method, was used to categorize the open text responses. The main purpose of content analysis is to capture the concepts and correlations that may explain the collected data (Şimşek & Yıldırım, 2011). Two members of the research team coded and classified the open-ended responses separately. Then, they collectively did this based on the analysis of the core concept for each aim. To ensure authenticity, investigator triangulation was used (Ma et al., 2009). A three-step coding approach was used to further refine the analysis of responses to the open-ended question.

A thematic analysis technique was used to identify, deduce and record trends in the responses (Clarke & Braun, 2013; Ritchie et al., 2013). A three-step coding approach was adopted (Gioia, Corley & Hamilton, 2013), similar to Strauss and Corbin’s open and axial coding (1990). The first step involved coding all the open-ended responses into several codes, then axial coding (Corbin & Strauss, 2014) was used to merge similar codes that were revealed during the first step, and cluster the codes into a less tangible form (Gioia, Corley & Hamilton, 2013). In the final step (Fig. 1) all similar codes that were found in the second step were grouped into three themes; (a) IMS perception of online teaching, (b) Barriers to online teaching, and (d) Future strategies for online teaching (Gioia, Corley & Hamilton, 2013) and the frequency of the codes was calculated (Fig. 2).

Figure 1 Representation of the coding tree that resulted from the coding process, with first-order codes, second-order codes, and aggregate themes.

Figure 2 Representation of the coding under three aggregate themes, the indication of the number of segments retrieved per code is shown as well.

Results

Demographics of participants

A total of 1,107 IMS participated in this study. Among them, 750 (67.8%) participants were males. Most (n = 698; 63.1%) were in the age range of 21 to 23 years, followed by 24 to 26 years (n = 204; 18.4%). The highest proportion of students were in their fourth year (n = 496; 44.8%), followed by third year (n = 248; 22.4%). Students from 12 countries participated in the study, the highest proportion being from Pakistan, 297 (26.8%), followed by Somalia 193 (17.4%), Indonesia 132 (11.9%) and India 111 (10%). For most of the students (n = 1,051; 94.9 %) their current location was their home country (Table 1).

Table 1 A table outlining the demographics (gender, age, year of program, country of origin and current location) of IMS responding to the survey (n = 1,107).

Variable	Frequency	Percentage (%)	
Gender			
Male	750	67.8	
Female	357	32.2	
Age			
18–20 years	153	13.8	
21–23 years	698	63.1	
24–26 years	204	18.4	
27 and above	52	4.7	
Year of program			
First-year	113	10.2	
Second-year	125	11.3	
Third-year	248	22.4	
Fourth-year	496	44.8	
Fifth-year	125	11.3	
Country of origin			
Pakistan	297	26.8	
Somalia	193	17.4	
Bangladesh	107	9.7	
Indonesia	132	11.9	
India	111	10	
Nigeria	81	7.3	
Sudan	51	4.6	
Ghana	43	3.9	
Kenya	39	3.5	
Tanzania	27	2.4	
Sri-Lanka	13	1.2	
Uganda	13	1.2	
Current location of IMS			
Home Country	1,051	94.9	
Other than the home country	5	0.5	
China	51	4.6	

Technology readiness of the study participants and general information about online classes

Most participants (n = 1,085; 98%), reported easy access to the internet. However, 664 (59.9%) participants rated their internet connection from poor to average, while 322 (29.9%) rated the quality of their connection as excellent. More than half of the participants (n = 581; 52.5%), reported impeded electrical supply. This is an alarming situation preventing Internet connection. Smartphone (n = 642; 58%) and laptop (n = 310; 28%) were selected by most participants as the preferred devices for accessing online classes. More than half (n = 587; 53%) of participants indicated that email was their preferred method of communication about the course schedule, followed by social media app WeChat 387 (35%). A total of 787 (71.1%) respondents reported attending three hours of online classes in a day. More than half of respondents (n = 659; 59.5%) reported that no assessment took place at the end of each class. A majority (n = 687; 62.1%) reported that three subjects were covered in a day (Table 2).

Table 2 A table displaying IMS choices on their technology readiness & information about online classes.

Statement	Response	Frequency & Percentage (%)	
Do you have easy access to the internet			
	Yes	1,085 (98%)	
	No	22 (02%)	
If yes, how would you grade your internet connectivity?
(1 = poor to 5 = excellent)			
	Poor	111 (10%)	
	Fair	333 (30.1%)	
	Average	220 (19.9%)	
	Good	121 (10.9%)	
	Excellent	322 (29.9%)	
Do you have an unimpeded electrical supply?			
	Yes	526 (47.5 %)	
	No	581 (52.5 %)	
Which of the following device do you use for online classes			
	Smartphone	642 (58%)	
	Laptop	310 (28%)	
	Tablet	122(11%)	
	Desktop	30 (2.7%)	
	Other	3 (0.3%)	
What is the mode of notification of the class schedule?			
	Via e-mail	587 (53%)	
	Via social media (e.g. WeChat, QQ)	387 (35%)	
	Via an Institution website	133 (12%)	
	Via SMS on cell phone	00 (0%)	
	Other	00 (0%)	
How long before the start of a class are you informed about the lecture schedule			
	One day before	365 (33%)	
	Two days before	609 (55%)	
	Few hours before	100 (09%)	
	One hour before	22 (02%)	
	Other	11 (01%)	
When did the online classes start after the 2020 vacations			
	March	804 (72.6 %)	
	April	289 (26.1 %)	
	May	14 (1.3 %)	
	June	00 (0%)	
	August	00 (0%)	
What is the duration of online teaching per day			
	1 h	00 (0%)	
	2 h	301 (27.2 %)	
	3 h	787 (71.1 %)	
	4 h	07 (0.6 %)	
	5 or more h	12 (1.1 %)	
Are you being assessed at the end of each class through a test or quiz?			
	Yes	448 (40.5 %)	
	No	659 (59.5 %)	
How many subjects are covered in one day?			
	One	00 (0%)	
	Two	401(36.2 %)	
	Three	687(62.1 %)	
	Four	07(0.6 %)	
	Five or more	12(1.1 %)	

IMS experience of online classes

Overall, students reported that online classes were not highly effective, and offered limited opportunities to interact with teachers. More than half (n = 614; 55.4%, 3.40 ± 1.40) of respondents agreed or strongly agreed with the statement that online classes hamper their attention and focus. Moreover, 801 (72.4%, 1.92 ± 1.24) of the respondents disagreed or strongly disagreed that online classes are equally or more informative than campus classes. Furthermore, 664 (60%, 1.83 ± 1.24) respondents strongly disagreed that online sessions should continue even after on-campus classes have restarted (Table 3).

Table 3 A table displaying students’ perceptions on their experiences of online teaching, ranked on a Likert scale from 1 to 5, where 1 = strongly disagree and 5 = strongly agree.

Likert scores have been shown as frequency, percentage and mean ± SD.

Statement	Strongly agree (n, %)	Agree (n, %)	Neutral (n, %)	Disagree (n, %)	Strongly disagree (n, %)	Mean ± SD	
My institution has an online learning management system (LMS) or Web site where all information about online classes is available	102 (9.2%)	349 (31.5%)	227 (20.5%)	238 (21.5%)	191 (17.3%)	2.93 ± 1.25	
All key information about the course is available on LMS or the institution Web site	136 (12.3%)	290 (26.2%)	221 (20%)	256 (23.1%)	204 (18.4%)	2.90 ± 1.30	
All course readings, assignments, and lectures are available online	136 (12.3%)	374 (33.8%)	221 (20%)	256 (23.1%)	120 (10.8%)	3.13 ± 1.21	
Students are assisted in overcoming obstacles in accessing the classes or materials	102 (9.2%)	323 (29.2%)	290 (26.2%)	290 (26.2%)	102 (9.2%)	3.02 ± 1.13	
Time allotted for online classes is sufficient	119 (10.7%)	392 (35.4%)	205 (18.5%)	323 (29.2%)	68 (6.1%)	3.15 ± 1.13	
I am able to interact with teachers during online classes	136 (12.3%)	374 (33.8%)	153 (13.8%)	239 (21.6%)	205 (18.5%)	2.99 ± 1.33	
I am able to interact with teachers after online class in the Q&A session	170 (15.4%)	255 (23%)	239 (21.6%)	239 (21.6%)	204 (18.4%)	2.95 ± 1.34	
Every individual is given a chance to participate and pitch in their ideas during online classes	136 (12.3%)	220 (19.9%)	256 (23.1%)	256 (23.1%)	239 (21.6%)	2.78 ± 1.31	
The teachers are well trained for online classes and are able to use the video conferencing app with ease	221 (20%)	170 (15.4%)	239 (21.6%)	170 (15.4%)	307(27.7%)	2.84 ± 1.48	
Attending classes from home hampers my attention and focus	307 (27.7%)	307 (27.7%)	187 (16.9%)	136 (12.3%)	170 (15.4%)	3.40 ± 1.40	
Online classes are equally or more informative as compared with active learning on campus	68 (6.1%)	85 (7.7%)	153 (13.8%)	187 (16.9%)	614 (55.5%)	1.92 ± 1.24	
Online learning fits in my schedules better than a typical day to day classes	34 (3.1%)	102 (9.2%)	170 (15.4%)	170 (15.4%)	631 (57%)	1.86 ± 1.16	
Demonstration of practical/clinical/lab work by the instructor during online classes would help me learn in a better way	85 (7.7%)	120 (10.8%)	136 (12.3%)	170 (15.4%)	596 (53.8%)	2.03 ± 1.33	
I would like to have these online sessions continued even after campus classes have started	68 (6.1%)	102 (9.2%)	68 (6.1%)	205 (18.5%)	664 (60%)	1.83 ± 1.24	
Note:

SD, standard deviation.

Comparison between public and private sector medical schools

A Mann–Whitney U test was conducted to compare satisfaction with online classes among students in public and private sector medical schools. As shown in Table 4, no statistically significant difference was found between IMS from public and private institutions. Students from both sectors were dissatisfied with various features of online teaching including: key information availability (Public 2.92 ± 1.30, Private 2.87 ± 1.32, P = 0.617), assistance received in overcoming difficulties (Public 3.04 ± 1.13, Private 2.99 ± 1.14, P = 0.603), teachers are well trained for online classes (Public 2.86 ± 1.47, Private 2.78 ± 1.50, P = 0.386), and whether attending classes from home hampers attention (Public 3.40 ± 1.39, Private 3.35 ± 1.42, P = 0.552).

Table 4 A Mann–Whitney U test results (p < 0.05): comparison of public and private sector medical institutes regarding the effectiveness of online teaching.

Survey questions	Public, n = 806	Private, n= 301	p-Value	
Mean ± SD	Mean ± SD	
My institution has an online learning management system (LMS) or Web site where all information about online classes is available	2.95 ± 1.25	2.91 ± 1.27	0.706a	
All key information about the course is available on LMS or the institution Web site	2.92 ± 1.30	2.87 ± 1.32	0.617a	
All course readings, assignments, and lectures are available online	3.14 ± 1.20	3.10 ± 1.23	0.649a	
Students are assisted in overcoming obstacles in accessing the classes or materials	3.04 ± 1.13	2.99 ± 1.14	0.603a	
Time allotted for online classes is sufficient	3.15 ± 1.14	3.14 ± 1.13	0.941a	
I am able to interact with teachers during online classes	2.99 ± 1.34	3.00 ± 1.32	0.950a	
I am able to interact with teachers after online class in the Q&A session	2.96 ± 1.33	2.91 ± 1.34	0.596a	
Every individual is given a chance to participate and pitch in their ideas during online classes	2.80 ± 1.31	2.72 ± 1.32	0.349a	
The teachers are well trained for online classes and are able to use the video conferencing app with ease	2.86 ± 1.47	2.78 ± 1.50	0.386a	
Attending classes from home hampers my attention and focus	3.40 ± 1.39	3.35 ± 1.42	0.552a	
Online classes are equally or more informative as compared with active learning on campus	1.92 ± 1.24	1.91 ± 1.24	0.878a	
Online learning fits in my schedules better than a typical day to day classes	1.87 ± 1.17	1.82 ± 1.13	0.544a	
Demonstration of practical/clinical/lab work by the instructor during online classes would help me learn in a better way	2.04 ± 1.34	1.99 ± 1.31	0.537a	
I would like to have these online sessions continued even after campus classes have started	1.85 ± 1.26	1.75 ± 1.19	0.193a	
Notes:

a p-Value calculated by using Mann–Whitney U test.

SD = standard deviation.

Effectiveness of online classes: an analysis of perceptions based on gender

A Mann-Whitney U test was used to examine IMS perceptions of effectiveness of the online classes. Table 5 shows that IMS were dissatisfied with the effectiveness of online classes as compared with active campus learning sessions, with no significant difference in dissatisfaction between genders (male, 1.89 ± 1.21, female, 1.96 ± 1.31, P = 0.613). Similarly, no significant gender difference was found in the view that online sessions should not continue after commencement of on campus classes (male, 1.81 ± 1.23, female, 1.85 ± 1.27, P = 0.785). These findings indicate that IMS perceive online medical education as unhelpful in several phases of the training, such as improving their clinical skills, knowledge and discussion skills.

Table 5 Comparison of male and female perceptions of online classes: a Mann–Whitney U test results (p < 0.05).

Survey questions	Male, n = 750	Female, n = 357	p-Value	
Mean ± SD	Mean ± SD	
My institution has an online learning management system (LMS) or Web site where all information about online classes is available	2.92 ± 1.28	2.97 ± 1.30	0.468a	
All key information about the course is available on LMS or the institution Web site	2.89 ± 1.28	2.93 ± 1.36	0.692a	
All course readings, assignments, and lectures are available online	3.12 ± 1.18	3.15 ± 1.26	0.587a	
Students are assisted in overcoming obstacles in accessing the classes or materials	3.01 ± 1.123	3.06 ± 1.15	0.565a	
Time allotted for online classes is sufficient	3.14 ± 1.11	3.17 ± 1.19	0.719a	
I am able to interact with teachers during online classes	2.97 ± 1.32	3.04 ± 1.37	0.401a	
I am able to interact with teachers after online class in the Q&A session	2.93 ± 1.32	2.98 ± 1.38	0.561a	
Every individual is given a chance to participate and pitch in their ideas during online classes	2.77 ± 1.28	2.79 ± 1.38	0.895a	
The teachers are well trained for online classes and are able to use the video conferencing app with ease	2.82 ± 1.45	2.87 ± 1.52	0.624a	
Attending classes from home hampers my attention and focus	3.38 ± 1.38	3.43 ± 1.44	0.402a	
Online classes are equally or more informative as compared with active learning on campus	1.89 ± 1.21	1.96 ± 1.31	0.613a	
Online learning fits in my schedules better than a typical day to day classes	1.84 ± 1.13	1.88 ± 1.21	0.837a	
Demonstration of practical/clinical/lab work by the instructor during online classes would help me learn in a better way	1.98 ± 1.29	2.12 ± 1.42	0.254a	
I would like to have these online sessions continued even after campus classes have started	1.81 ± 1.23	1.85 ± 1.27	0.785a	
Notes:

a p-Value calculated by using Mann–Whitney U test.

SD = standard deviation.

An analysis of perceptions based on year of study

A Kruskal–Wallis test was used to examine IMS perceptions regarding the effectiveness of online classes. Table 6 shows that dissatisfaction with the effectiveness of online classes (based on responses to questions such as ‘online learning fits in my schedules better than a typical day to day classes’) was similar in all five years of study (First year 1.84 ± 1.18; Second year 1.81 ± 1.13; Third year 1.77 ± 1.13; Fourth year 1.89 ± 1.16; Fifth year 1.93 ± 1.22; P = 0.613). These findings indicate that IMS in all phases of the program believe that online medical education is not fulfilling their needs.

Table 6 A Kruskal–Wallis test results (p < 0.05): year-wise comparison of IMS perceptions of online classes.

Survey questions	First year, n = 113	Second year, n = 125	Third year, n = 248	Fourth year, n = 496	Fifth year, n = 125	p-Value	
Mean ± SD	Mean ± SD	Mean ± SD	Mean ± SD	Mean ± SD	
My institution has an online learning management system (LMS) or Web site where all information about online classes is available	2.90 ± 1.26	2.85± 1.28	2.97± 1.24	2.93± 1.26	2.98 ± 1.28	0.882a	
All key information about the course is available on LMS or the institution Web site	2.87 ± 1.33	2.83 ± 1.34	2.94 ± 1.30	2.90 ± 1.29	2.96 ± 1.32	0.915a	
All course readings, assignments, and lectures are available online	3.11 ± 1.24	3.09 ± 1.25	3.14 ± 1.22	3.14 ± 1.19	3.15 ± 1.24	0.996a	
Students are assisted in overcoming obstacles in accessing the classes or materials	2.95± 1.18	2.96± 1.18	3.08± 1.16	3.03± 1.10	3.05 ± 1.12	0.812a	
Time allotted for online classes is sufficient	3.13 ± 1.16	3.11 ± 1.18	3.15 ± 1.12	3.16 ± 1.13	3.16 ± 1.15	0.994a	
I am able to interact with teachers during online classes	3.03 ± 1.31	2.96 ± 1.36	2.99 ± 1.35	3.00 ± 1.31	2.96 ± 1.40	0.995a	
I am able to interact with teachers after online class in the Q&A session	2.99 ± 1.36	2.97 ± 1.35	2.89 ± 1.36	2.95 ± 1.31	2.99 ± 1.37	0.946a	
Every individual is given a chance to participate and pitch in their ideas during online classes	2.72 ± 1.33	2.69 ± 1.34	2.74 ± 1.34	2.81 ± 1.29	2.85 ± 1.33	0.759a	
The teachers are well trained for online classes and are able to use the video conferencing app with ease	2.97 ± 1.46	2.85 ± 1.52	2.76 ± 1.52	2.84 ± 1.44	2.85 ± 1.50	0.793a	
Attending classes from home hampers my attention and focus	3.38 ± 1.38	3.46 ± 1.50	3.34 ± 1.39	3.41 ± 1.38	3.43 ± 1.38	0.854a	
Online classes are equally or more informative as compared with active learning on campus	1.88 ± 1.24	1.85 ± 1.21	1.84 ± 1.21	1.97 ± 1.25	1.96 ± 1.28	0.579a	
Online learning fits in my schedules better than a typical day to day classes	1.84 ± 1.18	1.81 ± 1.13	1.77 ± 1.13	1.89 ± 1.16	1.93 ± 1.22	0.621a	
Demonstration of practical/clinical/lab work by the instructor during online classes would help me learn in a better way	2.00 ± 1.35	1.98 ± 1.33	1.95 ± 1.32	2.08 ± 1.33	2.05 ± 1.35	0.647a	
I would like to have these online sessions continued even after campus classes have started	1.77 ± 1.23	1.79 ± 1.20	1.87 ± 1.26	1.82 ± 1.25	1.84 ± 1.26	0.930a	
Notes:

a p-Value calculated by using Kruskal–Wallis test.

SD, standard deviation.

IMS’ perspectives on the improvement of online teaching and future strategies

An optional open-ended question (‘‘Please tell us the three most crucial improvements required to make online sessions more effective and anything you want to share; please feel free to share’’) was addressed by 251 (22.67%) participants. A rigorous procedure was adopted for systematic coding. The qualitative data analysis method is described in Methods. See Fig. 2 for three main themes, related codes and their frequency.

Discussion

The COVID 19 pandemic forced educational institutions worldwide to adapt and implement online platforms for teaching (Akram et al., 2021; Wu et al., 2020). Here, we discuss how this situation has shaped the use of online teaching. Some of the findings were correlated with similar challenges in different environments; it is important to remember that our results reflect students’ diverse experiences from 12 countries. This investigation revealed dissatisfaction with online teaching among IMS enrolled with different medical institutions in China during the pandemic period. Previously, the courses included a rudimentary online medical education presence but the COVID-19 outbreak triggered much higher reliance on online teaching (Dost et al., 2020) and students became acquainted with various online learning methods and forums (Dhawan, 2020).

Experiences of IMS regarding the effectiveness of online teaching: technology readiness of IMS on uptake of online teaching

More than half of the respondents reported the quality of their internet connections as poor to average. Poor internet connection severely impacted students’ online learning experience. Persistent and recurring problems of internet access have emerged as a significant challenge faced by students and teachers, with previous studies reporting similar challenges to students’ online learning (Dridi et al., 2020). Family distractions, internet access, tutorial scheduling (Dost et al., 2020) and delayed communication (Howland & Moore, 2002; Vonderwell, 2003) are significant challenges in online teaching, and may disadvantage students with limited Internet access.

The current study also found that most students reported inadequate electrical supply during their virtual classes. This is a significant observation as it has been documented that the lack of sufficient electrical supply is one of the factors associated with tension among medical students, which may contribute to low academic success (Qamar, Khan & Bashir Kiani, 2015). Our findings confirm that electricity shortage and slow internet connections (common problems in rural areas) were perceived as leading barriers to course related information (Luqman et al., 2019). Power crises have been highlighted as contributing to reduced technology usage for access to education (Adil, Usman & Jalil, 2020). The academic literature is mostly silent on the relationship between the power crisis and online education.

The two most critical factors limiting access to online medical education programs may be power outages and poor Internet access, affecting students’ performance particularly in developing countries such as India, Tanzania, Pakistan, Somalia and others.

The present study found that respondents most commonly used smartphones to attend online classes, and the most popular means by which to receive class schedule notifications was social media (e.g., WeChat). Medical students have demonstrated positive attitudes toward learning with mobile technology (Abbasi et al., 2020; Hamilton et al., 2016; Suner, Yilmaz & Pişkin, 2019). Time management and convenient access to information are important reasons for medical students increasingly preferring to use smartphones (Bansal et al., 2020).

IMS perception of online teaching and challenges in adjusting to this new mode of learning

Students have reportedly expressed high levels of dissatisfaction with the institutional learning management system, the online availability of teaching information, their ability to engage with teachers during online classes, and the ability to overcome obstacles to access the classes and supplementary resources (Alenezi, 2018). Several authors have indicated that virtual education systems require a robust system of hardware and software that ensures reliable access to content and resources for successful learning (Asiry, 2017; Sarwar et al., 2020).

In the present study students also expressed general disappointment with teachers’ ability to offer online lectures effectively, an outcome of the need for rapid adaptation to online learning technology. This finding echoes those of other scholars who have argued that it is a demanding and time-consuming exercise to create persuasive, creative, and educational online content, and that this process requires a transitional preparation and adaptation period (Akram et al., 2021; Crawford et al., 2020; Sarwar et al., 2020)

In the present study, students acknowledged that their concentration and focus on learning are impeded by online classes. When asked whether online classes were more informative, most students disagreed. In addition, most students disagreed that online sessions should continue after regular campus operations have been completely resumed. These findings confirm those of previous studies in which the vast majority of students in dentistry and medicine appear to prefer traditional teaching methods and learning from textbooks and lectures (Abbasi et al., 2020; Bansal et al., 2020; Sarwar et al., 2020).

In contrast, some research indicates that students favor a more comprehensive pedagogical approach. To develop students’ understanding and expertise, traditional study sessions may be combined with online workshops including supplementary teaching materials and assignments (Dost et al., 2020; Hamilton et al., 2016). The development of groundbreaking educational initiatives to boost remote medical education has been initiated and may lead to success (Huddart et al., 2020).

Our findings indicate that students would like more collaborative online teaching lessons. This could be accomplished by integrating approaches such as polls, quizzes, or breakout rooms into student response systems (Dost et al., 2020; McBrien, Cheng & Jones, 2009), to promote student engagement (Morawo, Sun & Lowden, 2020). This active contact between teachers and students enables uncertainty to be quickly resolved to improve student engagement and build a more vibrant learning atmosphere.

Future directions and practical strategies for online teaching in medical institutes

Sufficient and updated resources contribute significantly to the online learning of students (Azevedo & Marques, 2017), and their proper channelization is crucial for successful accomplishment (Akram & Yang, 2021). IMS pointed out that compact pre-recorded videos would be helpful to learn remotely, previous research (Guo, Kim & Rubin, 2014) indicates the same. They further emphasized that the current pandemic climate needs a collective, harmonized, and global mutual effort to create successful online pedagogy practice policies. Moreover online education in developing countries remains a relatively recent concept, and strong efforts must be made to identify successful and efficient teaching strategies to address the barriers and obstacles preventing students’ access to effective online learning. Thus, the circumstances of the pandemic, lockdown and social distancing have impacted the learning of medical students.

Recommendations

The digitalization of medical teaching could play a significant role in the future of medical institutions (Dost et al., 2020). We believe that this critical student-facing issue should be addressed as soon as possible. Teachers should be encouraged to take part in online educational courses and programs that provide them with the tools they need to learn about different facets of immersive learning methods. This will help medical educators increase the effectiveness of existing online sessions. A smartphone-compatible virtual learning environment may be useful for online medical teaching. Problem-based learning or team-based learning has been found to improve learning outcomes (Clark, 2006; Yew & Goh, 2016), and student motivation and understanding (Chang, 2016). There is a need for a student support system to enhance the learning environment (Aslam et al., 2020). In a lockdown environment, medical institutions must launch external resources and training initiatives such as Osmosis and BiteMedicine. Universities need to immediately sign collaboration agreements with the medical institutions within students’ home countries to discuss teaching and learning arrangements for crisis periods.

Strengths, limitations and future research directions

To the researchers’ knowledge, this is the first study to examine IMS’ experiences of online teaching in China. The large sample size of 1,107 medical students from 12 countries from both pre-clinical and clinical years is one of the study’s strengths. In addition, the use of different approaches for the recruitment of medical students reduced possible bias in the responses. We studied the experiences of IMS in detail by analyzing their responses within groups based on the type of institution (public and private) as well as gender and year of study. We believe the results of our study provide important insights that will help medical institutions, educators and students to address and devise strategies to overcome the challenges to IMS learning posed by the COVID-19 pandemic.

The results may disproportionately represent views from students in some locations, with more responses from some countries (such as Pakistan) than others (such as Tanzania; Table 1), possibly introducing sampling bias. Another constraint of the study is that no assessment was made of whether institutions had trained their teachers professionally to conduct online classes, and the students’ response regarding teachers’ capacity to conduct classes online does not accurately reflect the teachers’ ability. When IMS interactions were explored in this research, no distinctions were made between various modes of online instruction. More in-depth observational surveys, such as focus groups/interviews/reflection journals, should be undertaken in collaboration with medical institutions to accurately measure the impact of COVID-19 on student use of online teaching. Future researchers should investigate alternatives to clinical sessions during any crisis requiring social distancing and how to conduct and control the quality of exams.

Conclusions

This study explored the perspectives of IMS regarding the effectiveness of online teaching, challenges in adjusting to this new mode of learning and proposed practical strategies for medical institutions based on the identified factors. IMS’ dissatisfaction with the various components of the online teaching indicates a need for medical institutions to enhance online learning. IMS suggested strategies including investment in faculty professional development and modification of online course content. More international collaboration may increase the quality and accessibility of online medical education. Virtual teaching, especially clinical simulation arrangements, should be developed collaboratively by advanced and developing countries, and would be helpful to IMS and assessment strategies. We recognize that COVID-19 has proven to be an extraordinary threat at the global level (Baloch et al., 2021) to which medical institutions have responded, but online education needs to be developed further. It will take time and experience to switch from a conventional face-to-face teaching approach to a fully functional virtual education framework in the medical education field.

Supplemental Information

Supplemental Information 1 SPSS data sheet.

Click here for additional data file.

Supplemental Information 2 Questionnaire used for data collection.

Click here for additional data file.

The authors would like to thank the Shaanxi Normal University Xi’an of China for technical support with data collection. The authors would also like to thank all the IMS for actively taking part in this study. Special thanks to Muhammad Mussab Umair for his assistance in distributing the survey link. We would also like to express our gratitude to Prof. Dr. Arshad Hasan, for his permission to use the questionnaire for this study.

Additional Information and Declarations

Competing Interests

Author Contributions

Human Ethics

Data Availability

The authors declare that they have no competing interests.

Sarfraz Aslam conceived and designed the experiments, performed the experiments, analyzed the data, prepared figures and/or tables, authored or reviewed drafts of the paper, and approved the final draft.

Huma Akram conceived and designed the experiments, performed the experiments, analyzed the data, prepared figures and/or tables, authored or reviewed drafts of the paper, and approved the final draft.

Atif Saleem conceived and designed the experiments, performed the experiments, analyzed the data, prepared figures and/or tables, authored or reviewed drafts of the paper, and approved the final draft.

BaoHui Zhang conceived and designed the experiments, performed the experiments, analyzed the data, authored or reviewed drafts of the paper, and approved the final draft.

The following information was supplied relating to ethical approvals (i.e., approving body and any reference numbers):

The Shaanxi Normal University granted Ethical approval to carry out the study within its facilities (Approval No: AR 2021-01-001).

The following information was supplied regarding data availability:

The raw measurements are available in the Supplemental Files.

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
