# Peer review of "Experiences of international medical students enrolled in Chinese medical institutions towards online teaching during the COVID-19 pandemic"

_PeerJ, doi:10.7717/peerj.12061_

## Round 0.1 · original submission · Major Revisions

Your manuscript has been reviewed and requires modifications before making a decision. The comments of the reviewers are included at the bottom of this letter. They also indicated that your manuscript needs English editing, and methods sections should be improved. I agree with this evaluation, and I would request the manuscript to be revised accordingly. I would also like to suggest the following changes:

The authors did not present any statistical analysis. Only frequencies were included in the manuscript. Statistical test results may improve the study results.

Please provide the version of the “IBM SPSS”.

What was the study power in this sample size?

Is the scale used in English or Chinese languages? Has the language validity of the scale been studied before? Does the scale have sub-dimensions and inverse items? If there are sub-dimensions, details should be shared for those dimensions as well.

Reviewer 1 ·

Basic reporting

The English writing could be improved. Somethings are not clear or need clarification.

There is sufficient field backround given.

I'm not sure that raw data is shared but it has a professional article structure.

The study states aims and these should be brought up again at the end.

I would like to know more about the themes and subthemes that you had as this does not appear in enough detail. I was expecting a figure about this but there was not one.

Experimental design

Research question pretty well defined but needs to be followed up in the discussion. It states how the article will fill a research gap. Investigation is performed to a good standard and ethical standard.

Methods need a bit more about the process of thematic analysis.

Validity of the findings

Describes how the article is novel. Data is quite robust and statistically sound.
Conclusions could link to the aims of the research more.

Additional comments

This is an interesting study. I have suggested some changes that may improve the work.
Sometimes the language flows and sometimes it doesn't flow very well ie the abstract is very well written but the introduction about Covid 19 etc is not as well written and could be more concise. Thank you for the opportunity to read your work.

Annotated reviews are not available for download in order to protect the identity of reviewers who chose to remain anonymous.

·

Basic reporting

1. It seems that word "responses" is not needed in the title.

Experimental design

1. Ln 135 "1107 IMS were recruited randomly studying in Chinese medical institutions."
Could you please explain in more details what random means, how very the respondents picked? From how many institutions were the students recruited?
2. Ln 143 questionnaire was in English I would suggest you state this fact.

Validity of the findings

Ln 328. It would be useful to give a short description of "Osmosis and BiteMedicine".
Lns 336.-338. "Some countries may have been disproportionately represented with more significant numbers of responses, such as Pakistan, possibly skewing outcomes because of sample bias compared to Tanzania" - are there any data or reference how IMS are from which country. If it is speculation, it should be identified as such.

Additional comments

A nice article on an interesting topic. It surely will be useful to improve online teaching.

---

## Round 0.2 · accepted · Accept

Thank you very much for the submission of a revised version of your paper. I have gone through the revised, track-changes manuscript and rebuttal letter, and see that the authors addressed the reviewers' concerns and substantially improved the content of the manuscript. So, based on my own assessment as an academic editor, the manuscript may be now accepted for publication.

Reviewer 1 ·

Basic reporting

English is MUCH improved. I can't fault it.
References provided.
Professional article structure.

Experimental design

No issues

Validity of the findings

No issues

Additional comments

I think this is ready for publication. The only thing I would say is there is a number like 456 thousand
and I would write 456,000 rather than use the word thousand - but extremely minor.

·

Basic reporting

The language has been significantly improved.

Experimental design

No comment.

Validity of the findings

No comment.